# A distinct tethering step is vital for vacuole membrane fusion

Michael Zick*, William T Wickner*

Department of Biochemistry, Geisel School of Medicine at Dartmouth, Hanover, United States

**Abstract** Past experiments with reconstituted proteoliposomes, employing assays that infer membrane fusion from fluorescent lipid dequenching, have suggested that vacuolar SNAREs alone suffice to catalyze membrane fusion in vitro. While we could replicate these results, we detected very little fusion with the more rigorous assay of lumenal compartment mixing. Exploring the discrepancies between lipid-dequenching and content-mixing assays, we surprisingly found that the disposition of the fluorescent lipids with respect to SNAREs had a striking effect. Without other proteins, the association of SNAREs in *trans* causes lipid dequenching that cannot be ascribed to fusion or hemifusion. Tethering of the SNARE-bearing proteoliposomes was required for efficient lumenal compartment mixing. While the physiological HOPS tethering complex caused a few-fold increase of *trans*-SNARE association, the rate of content mixing increased more than 100-fold. Thus tethering has a role in promoting membrane fusion that extends beyond simply increasing the amount of total *trans*-SNARE complex.

## Introduction

A hallmark of eukaryotic cells is their internal organization by distinct, highly specialized, membrane-enclosed compartments. Vesicular traffic functionally connects these individual compartments while preserving their integrity. The constant vectorial exchange between organelles is ensured by a complex machinery that facilitates the formation of vesicles with selective cargo, vesicle movement, specific tethering of each vesicle at its correct target membrane, and fusion of the vesicle and organelle membranes. Selective tethering and fusion are sequential events, but it remains unclear how tightly they are coupled. Tethering is commonly performed by specific proteins, often in large multi-subunit complexes that bind Rab family GTPases (**Stenmark, 2009**; **Yu and Hughson, 2010**). Fusion is mediated by soluble N-ethylmaleimide-sensitive attachment protein receptor (SNARE) proteins (**Jahn and Scheller, 2006**) and Sec1/Munc18 (SM) proteins (**Rizo and Sudhof, 2012**).

We study membrane fusion with the vacuoles (lysosomes) of *Saccharomyces cerevisiae* (**Wickner, 2010**). The vacuole is a large organelle that continuously undergoes fission and homotypic fusion with other vacuoles. At steady state, wild-type strains of yeast have just a few large vacuoles, but mutations which block vacuole fusion allow unimpeded fission, resulting in cells with numerous small vacuoles. This phenotype allowed the identification of the genes encoding vacuole fusion proteins (**Wada et al., 1992**). Vacuoles are readily isolable (**Bankaitis et al., 1986**), and a simple colorimetric assay of their fusion (**Haas et al., 1994**) permitted definition of the consecutive stages of fusion and the catalysts of each stage (**Ostrowicz et al., 2008**).

Vacuole tethering requires the Rab family GTPase Ypt7p and the heterohexameric Rab effector complex HOPS (homotypic vacuole fusion and protein sorting), which is comprised of Vps11p, Vps16p, Vps18p, Vps33p, Vps39p, and Vps41p (**Seals et al., 2000**; **Wurmser et al., 2000**). Two of these subunits, Vps39p and Vps41p, have direct affinity for Ypt7p, suggesting that one HOPS complex could tether two vacuoles, each bearing Ypt7p (**Brett et al., 2008**; **Ostrowicz et al., 2010**). Fusion of tethered membranes requires vacuolar SNAREs. Each vacuole has four SNAREs that are required for

**\*For correspondence:** michael.
zick@dartmouth.edu (MZ);
William.T.Wickner@dartmouth.
edu (WTW)

**Competing interests:** The authors declare that no competing interests exist.

**Reviewing editor**: Axel T Brunger, Stanford University, United States

**eLife digest** Cells of higher organisms contain compartments called organelles and structures called vesicles that transfer molecules and proteins between these organelles. Each organelle and each vesicle is enclosed within a membrane, and these membranes must fuse together to allow these transfers to take place. A certain group of proteins, called SNAREs, have a central role in these fusion events.

Since membrane fusion is difficult to observe directly, many researchers have used a method called 'fluorescent lipid dequenching' to study it indirectly. In this approach, one fraction of vesicles is labeled with two fluorescent molecules, with one of these molecules quenching the fluorescence of the other. However, when a labeled vesicle fuses with an unlabeled vesicle, the surface concentrations of the fluorescent molecules are diluted. This reduces the amount of quenching and the resulting increase in fluorescence can be measured.

Experiments utilizing this technique had suggested that SNARE proteins are sufficient for fusion to take place, and that no other protein complexes need to be present. However, when a different assay method called 'lumenal compartment mixing' was used, little fusion was seen when the only proteins present were the SNAREs. The lumenal compartment mixing approach relies on measuring the degree of mixing between the contents of two vesicles.

To address these conflicting results, Zick and Wickner used both methods to study fusion in a yeast-based system. The lumenal compartment mixing approach, which is the more reliable method, revealed that rapid and efficient membrane fusion in fact requires another protein complex, called HOPS, to hold the two membrane vesicles together.

Zick and Wickner found that the HOPS complex does not enable fusion by just increasing the amount of interactions between the SNARE proteins. Rather, it seems to facilitate the formation of a particular quality of SNARE interactions. Future work is needed to work out how the SNARE complexes become 'fusion-competent', and to explore the mechanism that allows the HOPS complex to assist in the formation of fusion-competent SNARE complexes.

---

homotypic fusion: the R-SNARE Nyv1p, the $Q_a$-SNARE Vam3p, the $Q_b$-SNARE Vti1p, and the $Q_c$-SNARE Vam7p (*Nichols et al., 1997*; *Ungermann et al., 1999*). Vam7p has the unusual property of lacking a hydrophobic membrane anchor while having an N-terminal membrane targeting domain with affinity for phosphatidylinositol 3-phosphate, HOPS, and acidic lipid (*Cheever et al., 2001*; *Lee et al., 2006*; *Stroupe et al., 2006*; *Karunakaran and Wickner, 2013*). Vacuolar *cis*-SNARE complexes are disassembled by Sec17p and Sec18p in an ATP-dependent manner, allowing the SNAREs to reassemble into *trans*-complexes, which are essential for fusion (*Haas and Wickner, 1996*; *Mayer et al., 1996*). HOPS fulfills several functions: Rab-dependent tethering (*Hickey and Wickner, 2010*), catalysis of the assembly of Vam7p into *trans*-SNARE complexes (*Zick and Wickner, 2013*), and protecting *trans*-SNARE complexes from Sec17p/Sec18p-mediated disassembly (*Xu et al., 2010*).

The reconstitution of vacuolar fusion in vitro has provided an important complementary approach for examining the individual steps of membrane fusion. In an early study, vacuole detergent extracts were reconstituted into proteoliposomes that could fuse with the purified organelle. Both the SNARE complex and vacuolar Rab GTPase Ypt7p were shown to be essential components of such detergent extracts (*Sato and Wickner, 1998*). Shortly thereafter, it was demonstrated that proteoliposomes bearing the vacuolar R-SNARE and two fluorescent lipids that exhibit fluorescence quenching, NBD-PE and rhodamine-PE, will interact with non-fluorescent proteoliposomes bearing the three Q-SNAREs to give fluorescence dequenching (*Fukuda et al., 2000*). This finding has since been confirmed in several additional studies (*Mima et al., 2008*; *Izawa et al., 2012*; *Furukawa and Mima, 2014*). A very recent study (*Furukawa and Mima, 2014*) has shown that vacuolar SNAREs are apparently unique among the yeast SNAREs, in that the purified SNAREs of the other yeast organelles will not mediate the dequenching of fluorescent lipids on proteoliposomes without additional proteins or tethering factors. In striking contrast, when fusion was assayed by the mixing of lumenally entrapped proteins while they remain inaccessible to an external competitor (*Zucchi and Zick, 2011*), very little fusion was seen without a tethering agent (*Zick and Wickner, 2013*). Thus, there are apparently contradictory reports as to whether SNAREs alone can efficiently mediate each stage of the fusion process.

We now report that vacuolar SNAREs alone will only support membrane fusion with an extremely low efficiency. Strikingly, fluorescent lipid dequenching can apparently occur due to *trans*-SNARE interactions that do not necessarily go on to fusion, emphasizing the pivotal importance of content mixing experiments in membrane fusion studies. A tethering/docking step is strictly required for rapid fusion, whether by the main physiological tethering system of Ypt7p and HOPS or by a vacuole-specific tethering mechanism mediated by the interaction of Vam7p's PX domain and PI(3)P in *trans*.

## Results

During biological membrane fusion, the barrier between the cytosol and the contents of the vesicle and target organelles remains intact while the lipids of two apposed bilayers rearrange to form one continuous membrane and the lumenal contents mix. Two principal types of assays have been employed to study this process in reconstituted systems, one based on the mixing of the bilayer lipids, the other based on the mixing of the vesicles' lumenal contents (*Figure 1A*). Lipid dequenching (*Figure 1A*, part 1) has been the most commonly used assay for membrane fusion. In this assay (*Struck et al., 1981*), one set of proteoliposomes bears two fluorescent lipids while the other has none. The two fluorophores are subject to fluorescence resonance energy transfer (FRET), so that the fluorescence emission of the 'donor' fluorophore is quenched due to its close proximity to the 'acceptor' fluorophore. FRET efficiency varies with the 6th power of the average distance between fluorophores (*Förster, 1948*); thus, fusion to unlabeled proteoliposomes, causing a twofold dilution of fluorescent lipids and thereby an increased average distance between fluorophores, results in a remarkable loss of quenching, and the ensuing increase in donor fluorescence is taken as a measure of fusion. In a modified version of this assay, donor and acceptor fluorophores start out on separate proteoliposome populations, and are brought into close contact when apposed bilayers fuse and their lipids mix (*Figure 1A*, part 2). The surrogate for fusion in this case is the decrease in fluorescence intensity of the donor fluorophore (due to quenching by the acceptor fluorophore via FRET). The quenching assay primarily reports the initial round of events, while the signal for the dequenching assay keeps increasing with consecutive rounds (*Figure 1—figure supplement 2*). However, hemifusion (a possible intermediate in which only the outer leaflets of apposed bilayers join) and lysis (disruption of the lipid bilayer with a loss of lumenal integrity) and ensuing re-annealing of bilayer fragments could also lead to lipid mixing which would be detected by these assays. Therefore, alternative assays that measure the mixing of lumenal contents in the continuous presence of an external competitor (*Figure 1A*, part 3), a strategy that has commonly been used to study the fusion of isolated organelles, are important as well.

We employ a reconstituted fusion assay (*Zucchi and Zick, 2011*) that allows us to measure both lipid and content mixing simultaneously. Proteoliposomes (*Figure 1B*) are prepared by dialysis of mixed micellar solutions of detergent, lipids which mimic the reported vacuolar composition (*Zinser and Daum, 1995*; *Schneiter et al., 1999*), two reporter fluorescent lipids (Marina-Blue-phosphatidylethanolamine (MB-PE) and/or NBD-PE), and pure recombinant proteins: the four vacuolar SNAREs, the prenylated Rab Ypt7p, and either of two lumenal markers (streptavidin derivatized with the fluorophore Cy5 [Sa-Cy5] or biotinylated R-phycoerythrin [PhycoE]). These proteoliposomes are mixed and incubated with HOPS, Sec17p, Sec18p, Mg$^{2+}$:ATP, and non-fluorescent streptavidin (to block any extra-lumenal association between biotinylated phycoerythrin and Cy5-streptavidin that might have been released by lysis). To measure both fusion and lysis, assays were also performed without added non-fluorescent streptavidin. In assays in which proteoliposomes bear each of the four SNAREs, as on the organelle itself, dequenching of the lipidic fluorophores (*Figure 1C*), lumenal compartment mixing which is continuously protected from external streptavidin (*Figure 1D*), and content mixing plus lysis (*Figure 1E*) required HOPS, Sec17p, and Sec18p (for this and subsequent figures, the extent of fusion after 10 min was used to estimate fusion activity; representative examples of kinetic data are shown as figure supplements).

Previous studies of organelle fusion (*Ungermann et al., 1998*), or dequenching experiments with proteoliposomes bearing SNARE combinations (*Fukuda et al., 2000*; *Mima et al., 2008*), have suggested that *trans*-SNARE complexes form between three Q-SNAREs anchored to one membrane and an R-SNARE (Nyv1p) anchored to the apposed membrane. Such an arrangement has allowed the study of sub-reactions with reduced complexity. We therefore prepared proteoliposomes as before (*Figure 1B*), but instead of bearing all four SNAREs, one fusion partner had only the R-SNARE Nyv1p and the other bore the 3Q-SNAREs. Surprisingly, these proteoliposomes showed very limited protected lumenal compartment mixing in the absence of other proteins (*Figure 1G*, column 1), although even

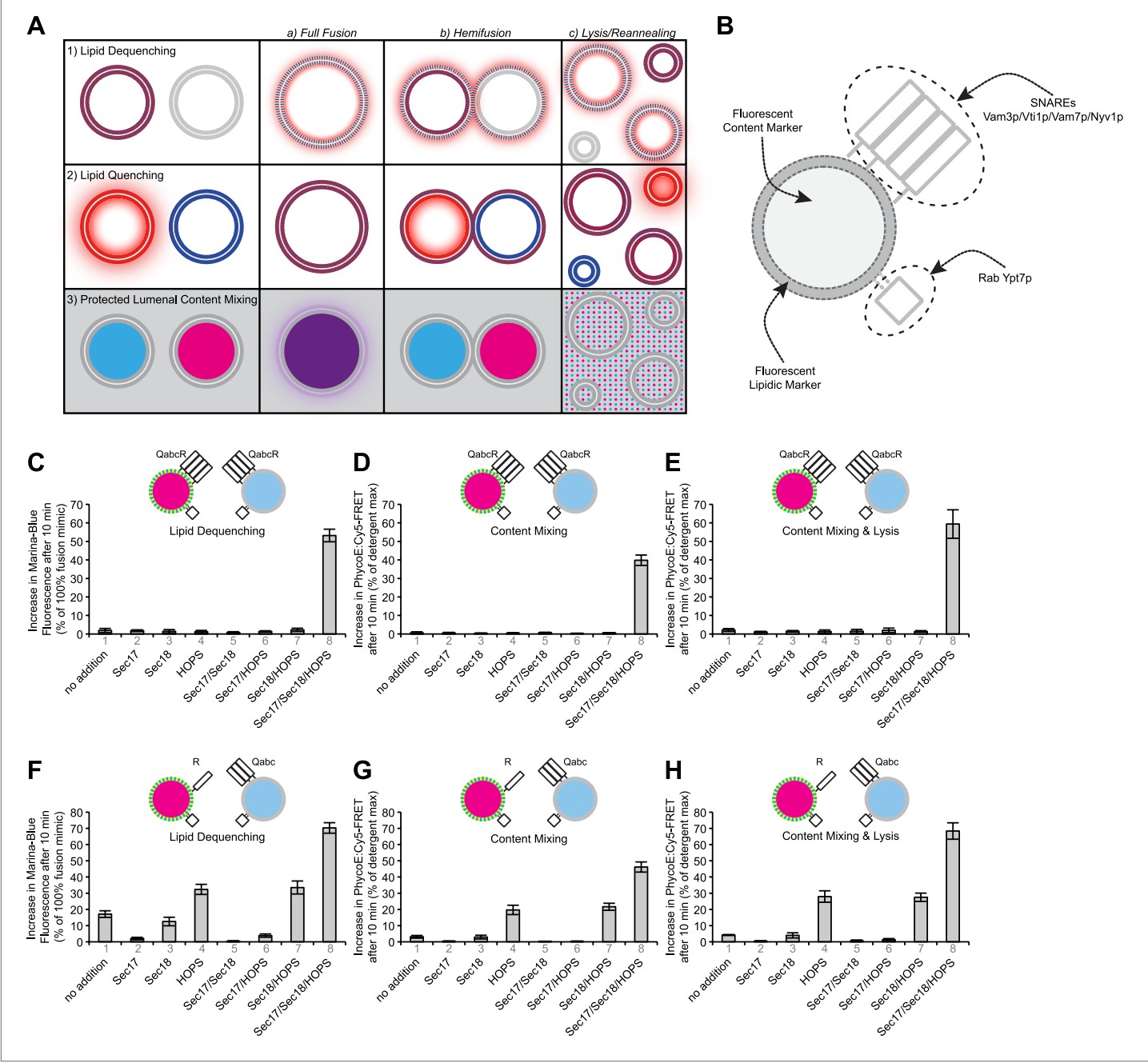

**Figure 1**. Comparison of lipid mixing and content mixing assays. (**A**) Assays used to study membrane fusion in vitro and possible outcomes. (**B**) Description of proteoliposomes as used in the following figures. (**C–E**) Two populations of 4SNARE-RPLs (with 1.5 mol% Marina-Blue-phosphatidylethanolamine [PE] and NBD-PE containing biotinylated R-phycoerythrin, or with no fluorescent lipids containing Cy5-labeled streptavidin; 250 µM lipid each) were incubated at 27°C with combinations of Sec17p, Sec18p, and HOPS, and assayed for 10 min for: lipid dequenching (**C**), lumenal content mixing protected from 5 µM external streptavidin (**D**), and mixing of lumenal content markers in the absence of external streptavidin, thus measuring both fusion and lysis (**E**). (**F–H**) Incubations were as in (**C–E**), but with RPLs bearing either the R-SNARE Nyv1p or the 3Q-SNAREs Vam3p, Vti1p, and Vam7p. Final concentrations of proteins were: Sec17p (600 nM), Sec18p (200 nM), and HOPS (100 nM). The mean and standard deviations of three experiments are shown.

The following figure supplements are available for figure 1:

**Figure supplement 1**. Representative kinetic data for panels C–H in *figure 1*.

**Figure supplement 2**. Sensitivity of lipid dequenching and lipid quenching assays to simulated rounds of fusion.

this very limited fusion was blocked by Sec17p/Sec18p (column 5) and thus likely required direct *trans*-SNARE interaction. Substantial fusion only occurred in the presence of HOPS and, as noted before (*Mima et al., 2008*), was further enhanced by Sec17p/Sec18p in addition to HOPS (columns 4 and 8). In contrast to the dependence of content mixing on HOPS, there was substantial lipid dequenching without the addition of HOPS (*Figure 1F*, column 1), as reported (*Mima et al., 2008*). To address this discrepancy, we turned to a more detailed analysis of the lipid dequenching assay.

## Lipid dequenching

Does the dequenching assay actually measure symmetric lipid mixing between the two proteoliposome populations? Proteoliposomes were prepared with both MB-PE and NBD-PE on the R-SNARE proteoliposomes or on the 3Q-SNARE proteoliposomes. Strikingly, the dequenching signal was seen when the fluorescent lipids were both on the R-SNARE proteoliposomes, but not when they were on the 3Q-SNARE proteoliposomes (*Figure 2A*, dark gray bars). For these direct comparisons of dequenching when the fluorophores are on one or the other proteoliposome, the proteoliposomes were mixed at a 1:1 ratio (*Figure 2A*). As in most reported dequenching assays, increasing the proportion of non-fluorescent acceptor 3Q-SNARE proteoliposomes enhances the signal such that up to 60% of maximal dequenching is achieved in 10 min (*Figure 2B*, light gray columns), whereas little dequenching of fluorescent 3Q-SNARE proteoliposomes is seen when incubated with even a 16-fold excess of non-fluorescent 1R-proteoliposomes (*Figure 2B*, dark gray columns). Providing an excess of 3Q-RPLs, but not 1R-RPLs, also promoted content mixing (see *Figure 5—figure supplement 2*) beyond what was seen at a 1:1 ratio of both RPLs, suggesting that only a fraction of the 3Q-RPLs are fusion competent. Dequenching does require *trans*-SNARE interactions: it is blocked by antibody to Vam3p, by recombinant soluble domain of Nyv1p, or by Sec17p/Sec18p (*Figure 2C*, light gray columns), which are

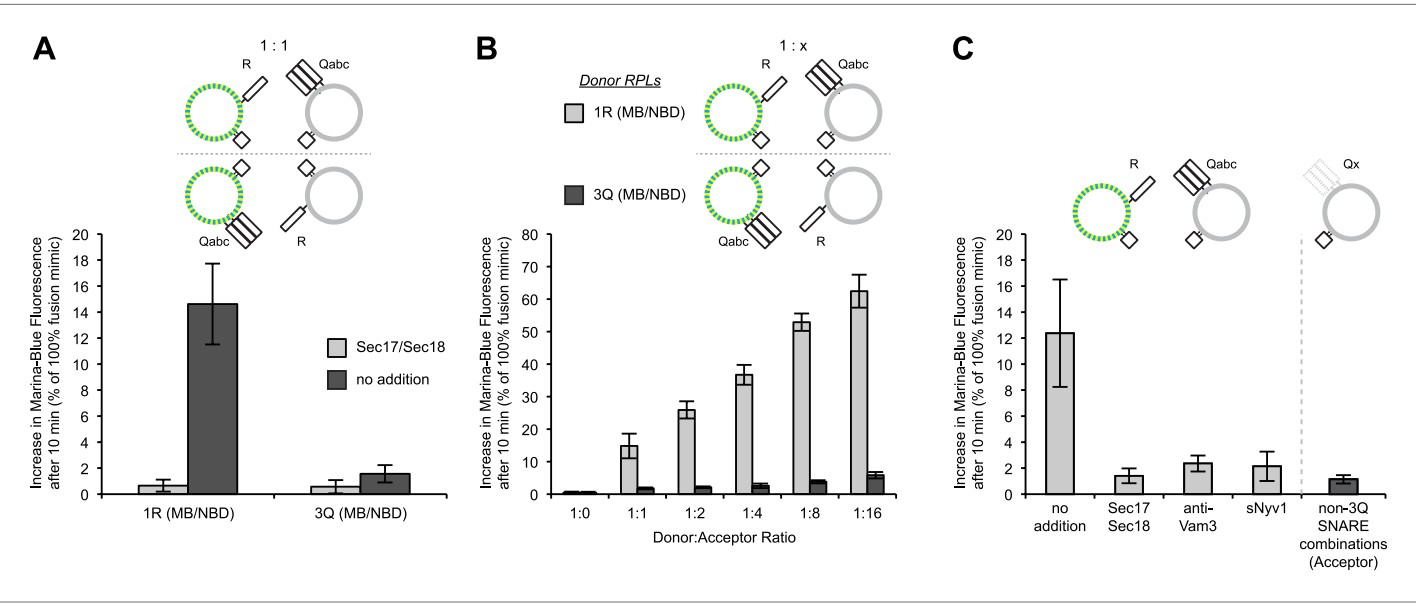

**Figure 2**. Asymmetric lipid dequenching. (**A**) Lipid dequenching reactions of 1R- and 3Q-RPLs (250 μM lipid each), with both fluorescent lipids (Marina-Blue-phosphatidylethanolamine [MB-PE] and NBD-PE, 1.5 mol% each) on either the 1R-RPLs or the 3Q-RPLs. Incubations contained either Sec17p (600 nM) and Sec18p (200 nM) (light gray) or no additional proteins (dark gray). (**B**) Lipid dequenching reactions of MB/NBD-labeled 1R- (light gray) or 3Q-RPLs (dark gray) with increasing amounts of non-labeled complementary RPLs (3Q or 1R; 1:1 corresponds to 50 μM lipid of each). No additional proteins were added. (**C**) Sensitivity of lipid dequenching reactions of 1R(MB/NBD)-RPLs and 3Q-RPLs to various inhibitors that might prevent *trans*-SNARE complex formation. Incubations received: (1) no addition, (2) Sec17p/Sec18p (400 nM each), (3) antibodies directed against Vam3p (1 μM), or (4) soluble Nyv1p(1-231) (1 μM). Column 5: 1R(MB/NBD)-RPLs were incubated with non-fluorescent RPLs bearing various combinations of SNAREs other than 3Q: $Q_{abc}R$, $Q_{bc}R$, $Q_{ac}R$, $Q_{ab}R$, $Q_cR$, $Q_bR$, $Q_{bc}$, $Q_aR$, $Q_{ac}$, $Q_{ab}$, R, $Q_c$, $Q_b$, $Q_a$. The extent of these reactions was comparably low for all conditions, and is shown as average for these 14 conditions with standard deviations.

The following figure supplement is available for figure 2:

**Figure supplement 1**. Representative kinetic data for panels A–C in *figure 2*.

reported (*Xu et al., 2010*) to disassemble *trans*-SNARE complexes. Each of the 14 compositions of SNAREs other than 3Q on the acceptor, non-fluorescent proteoliposomes did not support dequenching (*Figure 2C*, the average and standard deviation of these is shown for simplicity as a dark gray column). Thus, proteoliposomes can undergo *trans*-SNARE interactions that lead to asymmetric lipid dequenching without symmetrical lipid mixing or proportionate fusion.

Does the presence of the lipidic probes on the R-SNARE or 3Q-SNARE proteoliposomes affect the actual fusion reaction, as measured by protected lumenal content mixing? The asymmetry of lipid dequenching, controlled by the distribution of the probes (*Figure 3A*), is not reflected in asymmetry of fusion per se in the very same incubations (*Figure 3B*). Even lipid quenching assays, in which MB-PE and NBD-PE are initially incorporated separately into R-SNARE or 3Q-SNARE proteoliposomes, show an asymmetry in quenching (*Figure 3C*) that is not reflected in fusion per se (*Figure 3D*). This asymmetry could not be attributed to differences in RPL size (*Figure 3—figure supplement 2*), or fluorophore or protein incorporation (*Figure 3—figure supplement 3*).

Might dequenching that does not seem to represent fusion reflect a peculiar physical property of one or the other fluorophore? 3Q- and 1R-proteoliposomes were prepared with three sets of fluorescent PE probes, either MB and NBD, NBD and rhodamine, or Oregon-Green and Texas-Red. In the absence of HOPS or Sec17p/Sec18p, the same asymmetric dequenching signal was seen in each case (*Figure 4A–C*, no addition). We do not know how *trans*-SNARE interactions lead to this dequenching, but it is not symmetric, as expected for true lipid mixing, and is not proportionate to the content mixing for each reaction condition; thus, it cannot be relied on as a faithful reporter of fusion.

## A tethering requirement

While there was far less protected lumenal content mixing between 3Q- and 1R-SNARE proteoliposomes in the absence of HOPS, Sec17p, and Sec18p than in their presence (*Figure 1G*, columns 1 vs 8), even this low level (column 1) was suppressed by Sec17p/Sec18p (column 5) and thus presumably relied on the SNAREs. Was this fusion independent of tethering, that is, did it rely on direct formation of stable proteoliposome adherence through 4-SNARE coiled-coil domain assembly, or did another kind of interaction mediate tethering? The Vam7p N-terminal PX domain has direct affinity for PI(3)P (*Cheever et al., 2001*), which is important for fusion (*Fratti and Wickner, 2007*). 3Q- and 1R-proteoliposomes were prepared with or without PI(3)P. They were incubated in each combination without further protein addition, with HOPS, with Sec17p/Sec18p, or with HOPS/Sec17p/Sec18p, and their protected lumenal compartment mixing was assayed (*Figure 5A*). Strikingly, with PI(3)P only on the R-SNARE proteoliposomes, fusion was as vigorous without HOPS as in its presence (*Figure 5*, columns 5 and 7). This fusion was blocked by Sec17p/Sec18p (column 6) and was not seen when the PI(3)P was entirely absent (column 1) or when PI(3)P was only on the 3Q-SNARE proteoliposomes (column 9). The content mixing that was seen when PI(3)P was on the R-SNARE proteoliposomes alone was strongly suppressed when PI(3)P was present on the 3Q-RPLs as well (columns 5 vs 13), presumably because the PX domain is engaged in an interaction with PI(3)P in *cis*, rendering a *trans* interaction less likely. The delay in reaction kinetics for conditions with Sec17, Sec18, and HOPS and no PI(3)P on the 3Q-RPLs (*Figure 5—figure supplement 1A,B*, purple lines) probably reflects the requirement for phosphoinositides for the synergy between Sec17/Sec18 and HOPS (*Mima and Wickner, 2009*). The fusion that is supported by the presence of PI(3)P on the R-SNARE proteoliposomes can be ascribed to its binding in *trans* by the PX domain of Vam7p, as 3Q-SNARE proteoliposomes which were made with Vam7pΔPX will not fuse with R-SNARE proteoliposomes which bear PI(3)P (column 17). After 10 min assays, the very low lumenal compartment mixing values observed with the RPLs alone were not significantly above those seen with inhibition by Sec17p and Sec18p (*Figure 5A*, lower panel, columns 1 and 2). To determine whether SNARE-bearing RPLs alone can undergo any full fusion at all on their own, we performed an extended fusion assay with 3Q- and 1R-proteoliposomes without PI(3)P in the presence or absence of added HOPS (*Figure 5B*). Detectable fusion could in fact be supported by SNAREs alone (blue), albeit at less than a 100th the rate as seen in the presence of HOPS (red).

## Fusion and *trans*-association of SNAREs

Does this dramatic difference in fusion rates reflect a proportionate inability of the three Q-SNAREs on proteoliposomes to engage with R-SNAREs in the absence of a tether? To study how much *trans*-interaction occurs between membrane-bound 3Q-SNAREs and membrane-bound R-SNAREs in the presence or absence of HOPS, we performed a content mixing reaction (*Figure 6A*) and also subjected

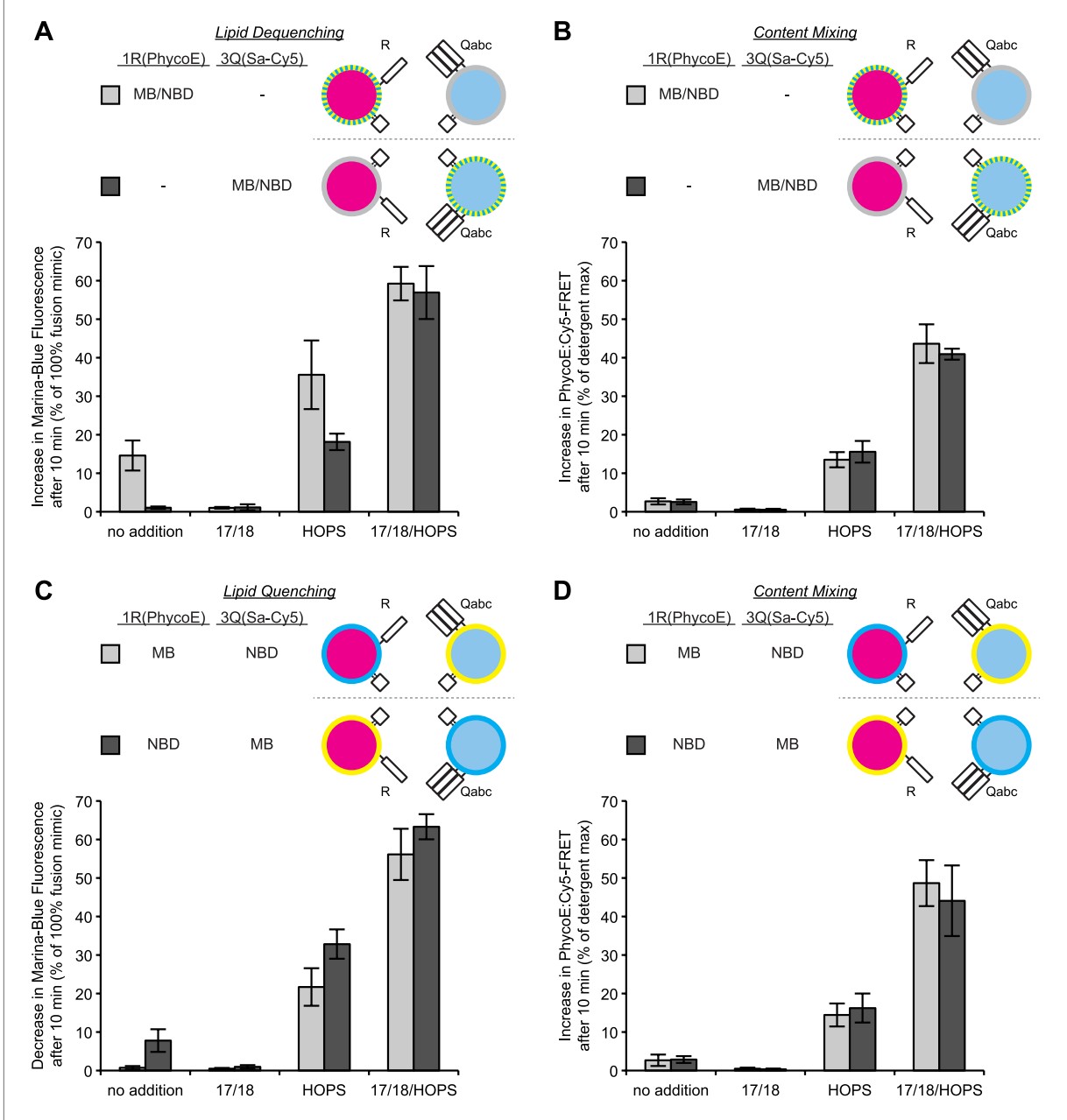

**Figure 3**. Discrepancies between lipid mixing and content mixing assays in 1R-3Q RPL fusion reactions. Incubations contained Sec17p (600 nM), Sec18p (200 nM), and HOPS (100 nM), as indicated. (**A** and **B**) Incubations contained 1R-RPLs (containing Biotin-PhycoE) and 3Q-RPLs (containing Streptavidin-Cy5) (250 μM lipid each), with the two fluorescent lipids Marina-Blue-phosphatidylethanolamine (MB-PE) and NBD-PE (1.5 mol% each) being together on either the 1R-RPLs or the 3Q-RPLs. Lipid dequenching (**A**) and protected lumenal content mixing (**B**) were recorded from the same reactions. (**C** and **D**) Incubations bore 1R-RPLs (containing Biotin-PhycoE) and 3Q-RPLs (containing Streptavidin-Cy5) (250 μM lipid each), with each RPL bearing a single fluorescent lipid, either Marina-Blue-phosphatidylethanolamine (MB-PE) (1 mol%) or NBD-PE (3 mol%). Lipid quenching (**C**) and protected lumenal content mixing (**D**) were recorded from the same reactions.

The following figure supplements are available for figure 3:

**Figure supplement 1**. Representative kinetic data for panels A–D in *figure 3*.

**Figure supplement 2**. Characterization of liposome size by dynamic light scattering.

**Figure supplement 3**. Characterization of dye and protein incorporation.

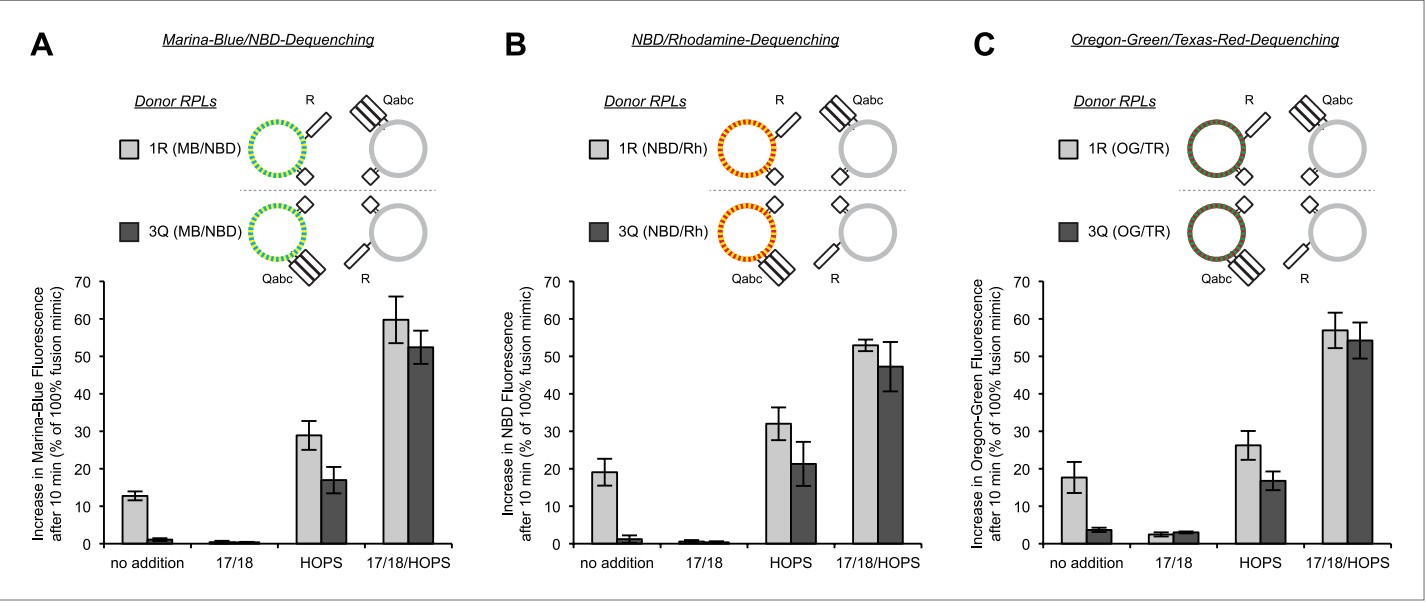

**Figure 4**. Asymmetric lipid dequenching is seen with different pairs of fluorescent lipidic markers. (**A–C**) Lipid dequenching reactions of 1R- and 3Q-RPLs (250 µM lipid each) had combinations of Sec17p (600 nM), Sec18p (200 nM), and HOPS (100 nM), as indicated. Either the 1R-RPLs (light gray) or the 3Q-RPLs (dark gray) bore pairs of fluorescently labeled lipids (1.5 mol% each): Marina-Blue (MB) and NBD (**A**), NBD and rhodamine (Rh) (**B**), or Oregon-Green (OG) and Texas-Red (TR) (**C**).

The following figure supplement is available for figure 4:

**Figure supplement 1**. Representative kinetic data for panels A–C in *figure 4*.

samples which had been detergent-solubilized to immunoprecipitation with anti-Vam3p antibodies to determine the amount of Nyv1p that co-precipitated as a measure of *trans*-SNARE interaction (*Figure 6B*). The detergent bore a large excess of GST-Nyv1p, sufficient to suppress any association between the 3Q-SNAREs and wild-type Nyv1p after detergent-solubilization (*Figure 6*, *Figure 6—figure supplement 1*). The presence of HOPS led to an increase in the *trans*-association of membrane anchored SNAREs (*Figure 6B*, compare lanes 3 and 4). However, this increase was less than fourfold, and thus insufficient to explain the difference in content mixing rates (*Figure 6A*). Quantitation of replicate experiments (*Figure 6C*) shows that about one third as much *trans*-SNARE complex formed in the absence of HOPS (column 3) as in its presence (column 4), whereas fusion was over 100 times faster in the presence of HOPS (columns 1, 2). Might this reflect an inability of the 3Q-SNAREs to interact with Nyv1p without HOPS? When incubations bore the recombinant soluble domain of Nyv1p at the same 100 nM concentration as was otherwise present on RPLs in our assay, the soluble Nyv1p readily bound to the Q-SNAREs regardless of the presence or absence of HOPS (*Figure 6B*, compare lanes 7 and 8). Furthermore, 100 nM soluble Nyv1p can suppress most of the content mixing reaction (*Figure 6D*), just as fusion was blocked by the mixed soluble domains of the three Q-SNAREs (*Figure 6—figure supplement 3*), and inhibition by added soluble domain of Nyv1p was reversible by including the SNARE complex disassembly chaperones Sec17/Sec18 (*Figure 6E*). This indicates that the 3Q-SNARE complexes were capable of engaging in complex formation with membrane-bound R-SNAREs, but in a manner that did not lead to fusion. To test whether the highly disproportional increase in *trans*-SNARE association and content mixing can be attributed to a possible requirement for high cooperativity of SNARE complexes, we examined the levels of content mixing and *trans*-SNARE interactions when R-RPLs were used that had been reconstituted with lower amounts of Nyv1p (*Figure 7*). Lower levels of Nyv1p did not lead to a drastic diminution of fusion activity (*Figure 7A,C*), but resulted in lower amounts of *trans*-SNARE association (*Figure 7B,D*). Since the amount of *trans*-SNARE interaction that was detected at low Nyv1p levels with HOPS was lower than the amount seen with high Nyv1p levels without HOPS (*Figure 7B,D*, lane 2 vs lane 7), yet the amount of fusion was drastically higher in the sample containing HOPS (*Figure 7A,C*, lane 2 vs lane 7), the requirement for high

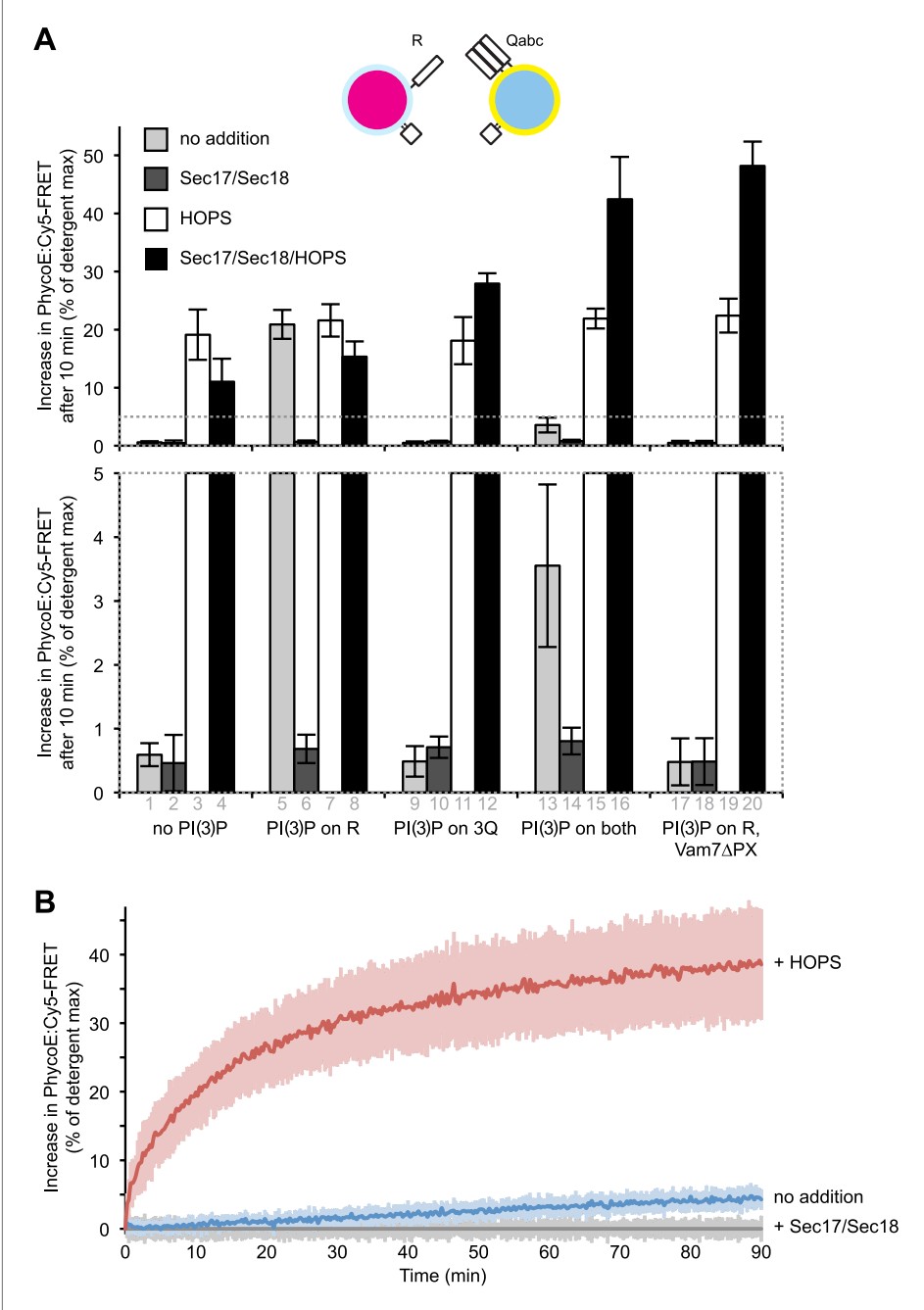

**Figure 5**. Modulation of fusion activity through asymmetric distribution of PI(3)P. (**A**) 1R-RPLs (with 0.5 mol% Marina-Blue-phosphatidylethanolamine [PE], containing biotinylated R-phycoerythrin) and 3Q-RPLs (with 3 mol% NBD-PE, containing Cy5-labeled streptavidin), bearing PI(3)P as indicated, were assayed for protected lumenal compartment mixing. Where indicated (columns 17–20), 3Q-RPLs bore a truncated version of the Qc-SNARE Vam7p that lacks the PI(3)P binding PX domain (**Xu and Wickner, 2012**). Reactions contained Sec17p (600 nM), Sec18p (200 nM), and HOPS (100 nM), as indicated. The lower panel provides a magnified view of the bottom 5% of the data (indicated by dotted box). (**B**) Kinetics of 1R-3Q content mixing assays with RPLs not bearing PI(3)P. Reactions contained HOPS (100 nM; red curve), Sec17p and Sec18p (600 and 200 nM; gray curve), or no added proteins (blue curve). The averages of five experiments are shown. The reactions containing Sec17p/Sec18p were normalized to zero, and these values were subtracted from all other conditions. Light-colored areas indicate standard deviations.

*Figure 5. Continued on next page*

*Figure 5. Continued*

The following figure supplements are available for figure 5:

**Figure supplement 1**. Representative kinetic data for panel A in **figure 5**.

**Figure supplement 2**. Effect of varying the ratio of 1R to 3Q RPLs.

cooperativity can be ruled out as an explanation for the observed discrepancy between increases in *trans*-SNARE interaction and fusion.

## Discussion

Our knowledge of the initial steps of membrane fusion is limited, and the terminology to describe these processes is vague. 'Tethering' often refers to *trans*-interactions of membranes that are of low affinity, reversible, or do not involve SNAREs, and 'docking' is reached when *trans*-SNARE complexes have formed. These terms are used loosely, even interchangeably, and it has remained unclear whether and how the functions of Rab GTPases, their effectors, SM proteins, and SNAREs are coupled or even related. The steps that lead to *trans*-SNARE complex formation are dynamic and interwoven, and we are only beginning to understand how they lead to membrane fusion.

Several biochemical assays have been developed to monitor the fusion process in vitro. Most studies with isolated organelles have assayed fusion as the mixing of lumenal contents. Studies with reconstituted proteoliposomes, in contrast, have relied heavily on lipid dequenching assays to infer the merging of two lipid bilayers. Discrepancies of lipid mixing and content mixing have repeatedly been noted in the past: for example, neuronal SNAREs alone do not produce much content mixing yet readily promote lipid dequenching (*Kyoung et al., 2011*; *Diao et al., 2012*), content mixing occurs seconds after initial lipid mixing in influenza virus-induced fusion (*Floyd et al., 2008*), and content mixing occurs with a delay of several minutes after lipid mixing in vacuolar fusion (*Jun and Wickner, 2007*).

We have recently described an assay that allows concurrent measurement of both lipid and content mixing (*Zucchi and Zick, 2011*). We have reconstituted a fusion reaction that mimics the homotypic fusion of yeast vacuoles, that is, with the four SNAREs in a *cis*-complex on both membranes, depending on Sec17p/Sec18p for ATP-dependent priming and Ypt7p:HOPS for tethering (see *Wickner, 2010* for review). Both content and lipid mixing assays show that this reaction requires Sec17p, Sec18p, and HOPS (*Figure 1C,D*). A modified reaction scheme, in which one population of RPLs bears the R-SNARE Nyv1p, while the other population bears the three Q-SNAREs Vam3p, Vti1p, and Vam7p, allows study of a sub-reaction that bypasses the need for Sec17p/Sec18p mediated priming. This condition has repeatedly been reported to allow lipid mixing to occur without the need for any additional components (*Fukuda et al., 2000*; *Mima et al., 2008*). While we could reproduce this finding (*Figure 1F*, column 1), we observed a substantially lower amount of content mixing for the same condition (*Figure 1G*, column 1). Furthermore, among the several sets of SNAREs of yeast organelles, only the vacuolar SNAREs showed a purported fusion signal (*Izawa et al., 2012*; *Furukawa and Mima, 2014*). Lipid mixing without proportional content mixing may indicate hemifusion (see *Figure 1A*). However, the lipid dequenching signal showed an asymmetric character, depending on the two fluorescent lipids being present on the R-SNARE-bearing RPLs (*Figure 2A,B*). It is nevertheless possible to suppress this signal with well-established inhibitors of fusion that interfere with the formation of *trans*-SNARE complexes (*Figure 2C*). The fluorescent lipids did not appear to modulate the capacity of RPLs to fuse, as content mixing was not affected by the presence or distribution of fluorescent lipids (*Figure 3*). The effect was also not limited to a specific set of fluorescent lipids (*Figure 4*). We cannot explain this asymmetric behavior, but it may be related to an undefined interaction between vacuolar SNAREs and the fluorescent lipids. The latter might be selectively transferred to another membrane that contains a particular SNARE via an interaction with that SNARE; transfer of lipid dyes between membranes has been reported for rhodamine B lipids (*Ohki et al., 1998*). Basic residues in the juxtamembrane regions of a Q-SNARE might facilitate transfer of the fluorescent lipids from the R-RPLs when the two membranes are brought into proximity by *trans*-SNARE complex assembly, thus catalyzing transfer from the R- to the 3Q-RPLs. In any case, the lipid mixing assay needs to be employed with great caution, and should ideally only serve as a complement

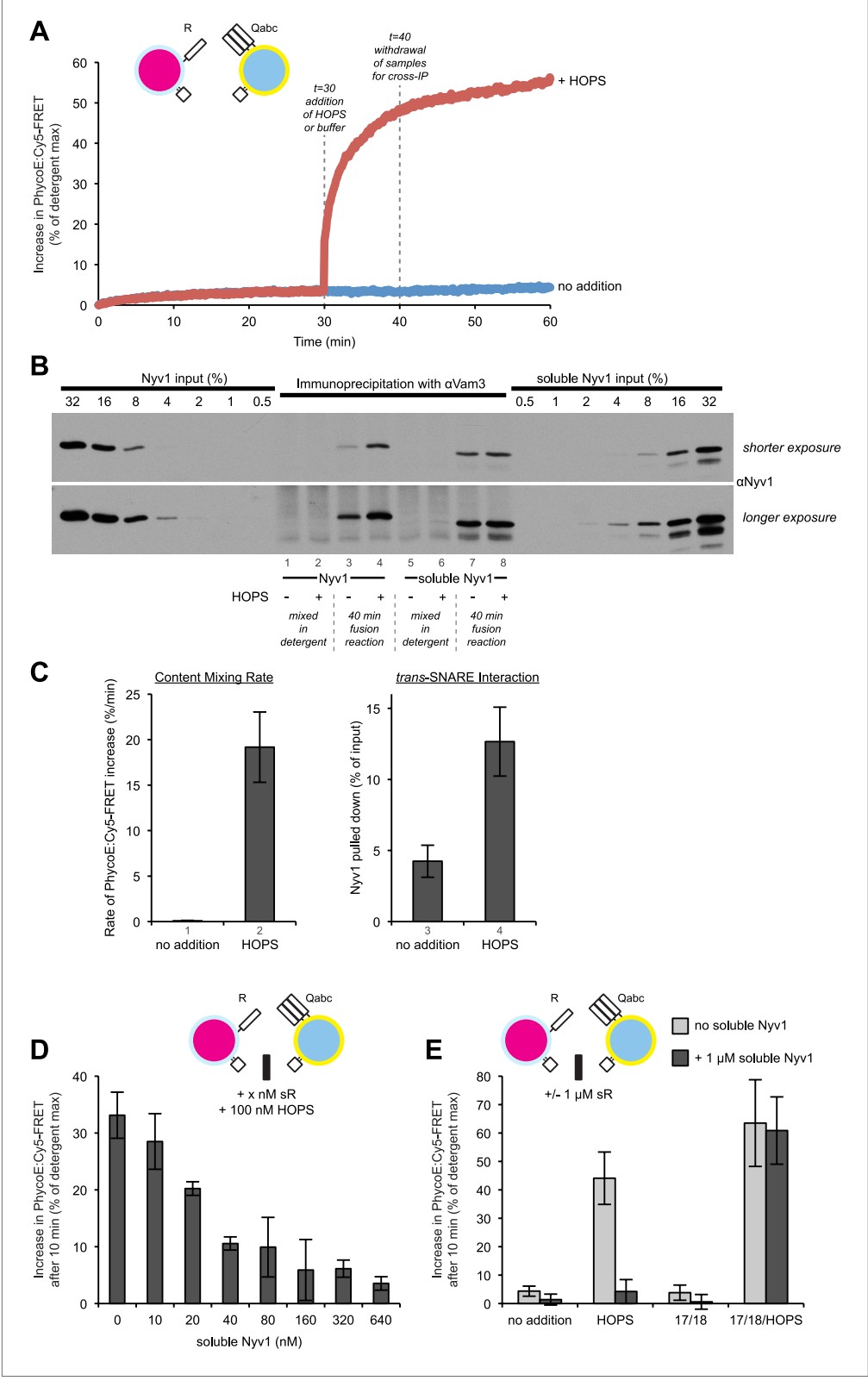

**Figure 6**. HOPS increases the rate of content mixing substantially more than the formation of *trans*-SNARE associations. (**A**) Kinetics of 1R-3Q content mixing assays. RPLs (reconstituted with a protein:lipid ratio of 1:2500 for SNAREs) were incubated for 30 min at 27°C before and after the addition of HOPS (100 nM) or its buffer. *Figure 6. Continued on next page*

*Figure 6. Continued*

(**B**) Additional 40 µl reactions that were performed in parallel were analyzed for *trans*-SNARE association by co-immunoprecipitation of Nyv1p (R-SNARE) with Vam3p ($Q_a$-SNARE). Lanes 3 and 4 correspond to the conditions as shown in (**A**). Lanes 7 and 8 are from equivalent reactions, but containing soluble Nyv1p instead of R-RPLs. For lanes 1, 2, 5, and 6, the components were mixed directly in detergent solution without prior incubation. (**C**) Quantification of content mixing and *trans*-SNARE association (as seen in panels **A** and **B**) from three independent repeat experiments. The rate of fusion was determined as the slope for the first minute of the content mixing reaction after addition of HOPS or its buffer (between minutes 30 and 31). *Trans*-SNARE interactions were quantified as the amount of Nyv1p that co-precipitated with Vam3p (lanes 3 and 4 in panel **B**) using UN-SCAN-IT gel 5.3 software (Silk Scientific, Orem, UT). (**D**) 1R- and 3Q-RPLs were mixed with increasing concentrations of soluble Nyv1p (sR), and incubated for 30 min at 27°C. HOPS (100 nM) was added, and the reactions were assayed for protected lumenal compartment mixing. (**E**) A mixture of 1R- and 3Q-RPLs was pre-incubated with or without 1 µM soluble Nyv1p for 30 min at 27°C. Sec17p (600 nM), Sec18p (200 nM), and HOPS (100 nM) or their respective buffers were added as indicated, and the reactions were assayed for protected lumenal compartment mixing.

The following figure supplements are available for figure 6:

**Figure supplement 1**. Suppression of the formation of 1R-3Q SNARE complexes in detergent by increasing concentrations of GST-tagged Nyv1p.

**Figure supplement 2**. Representative kinetic data for panels D–E in *figure 6*.

**Figure supplement 3**. Reversible inhibition by soluble Q-SNAREs.

to content mixing experiments. Lipid dequenching has provided many valuable lessons, but can apparently report events other than true fusion.

A vigorous content mixing signal, which is always accompanied by a lipid mixing signal (dequenching or quenching), is only seen in the presence of HOPS (*Figure 3*), indicating that tethering is a vital component of the fusion process. The small amount of content mixing in 1R-3Q reactions that was seen without added protein was suppressed by the addition of Sec17p/Sec18p (*Figure 1G*, compare columns 1 and 5), indicating that it did indeed represent SNARE-dependent true membrane fusion. A previous report has shown that the *trans*-interaction of PI(3)P and the PX domain of the Qc-SNARE Vam7p could promote tethering (*Xu and Wickner, 2010*). This interaction was partially responsible for the SNARE-dependent content mixing seen without additional proteins, since the complete absence of PI(3)P, its asymmetric disposition on 3Q-RPLs, or a truncated form of Vam7p lacking its PX domain further reduced the small amount of content mixing seen without HOPS (*Figure 5A*). In contrast, when PI(3)P was only present on R-SNARE RPLs, rapid fusion did not require HOPS (*Figure 5A*, columns 5 and 6), establishing a condition with reduced complexity in which an SM function is not absolutely required. It remains to be determined whether the Vam7p:PI(3)P tethering plays any role on intact vacuoles, where priming releases Vam7p early in the fusion reaction (*Boeddinghaus et al., 2002*), or if it is merely a phenomenon seen in in vitro reconstitutions with purified components. In this context, we note that the content mixing supported by Vam7p:PI(3)P tethering is fully suppressed in the presence of Sec17p/Sec18p (*Figure 5A*, column 6). The ubiquitous presence of Sec17p/Sec18p in the cell may underlie a strict requirement for HOPS and other large tethering complexes in vivo. HOPS has the capacity to recruit Vam7p (*Zick and Wickner, 2013*), to proofread *trans*-SNARE complexes (*Starai et al., 2008*), and to protect them from disassembly by Sec17p/Sec18p (*Hickey and Wickner, 2010*; *Xu et al., 2010*).

While the requirement for a tether was not absolute, the addition of HOPS stimulated the rate of the reaction more than 100-fold (*Figure 5B*, *Figure 6C*). Colliding SNARE-bearing membranes form spontaneous *trans*-SNARE complexes (*Figure 6D*, lane 3). These can occasionally trigger fusion, but the rate achieved under such a condition is of questionable physiological relevance. This uncertainty is reinforced by the fact that mutations in Rab GTPases or tethering complexes have as strong a phenotype as mutations in SNARE proteins (*Wada et al., 1992*). The addition of HOPS stimulates the fusion reaction more than 100-fold, yet it only increases the amount of *trans*-SNARE interaction a few-fold (*Figure 6C,D*). This shows that the role of HOPS and tethering goes beyond merely facilitating *trans*-SNARE complex formation. The *trans*-SNARE complexes that form spontaneously are very inefficient in catalyzing membrane fusion. This indicates that it is not just a matter of quantity, but that additional

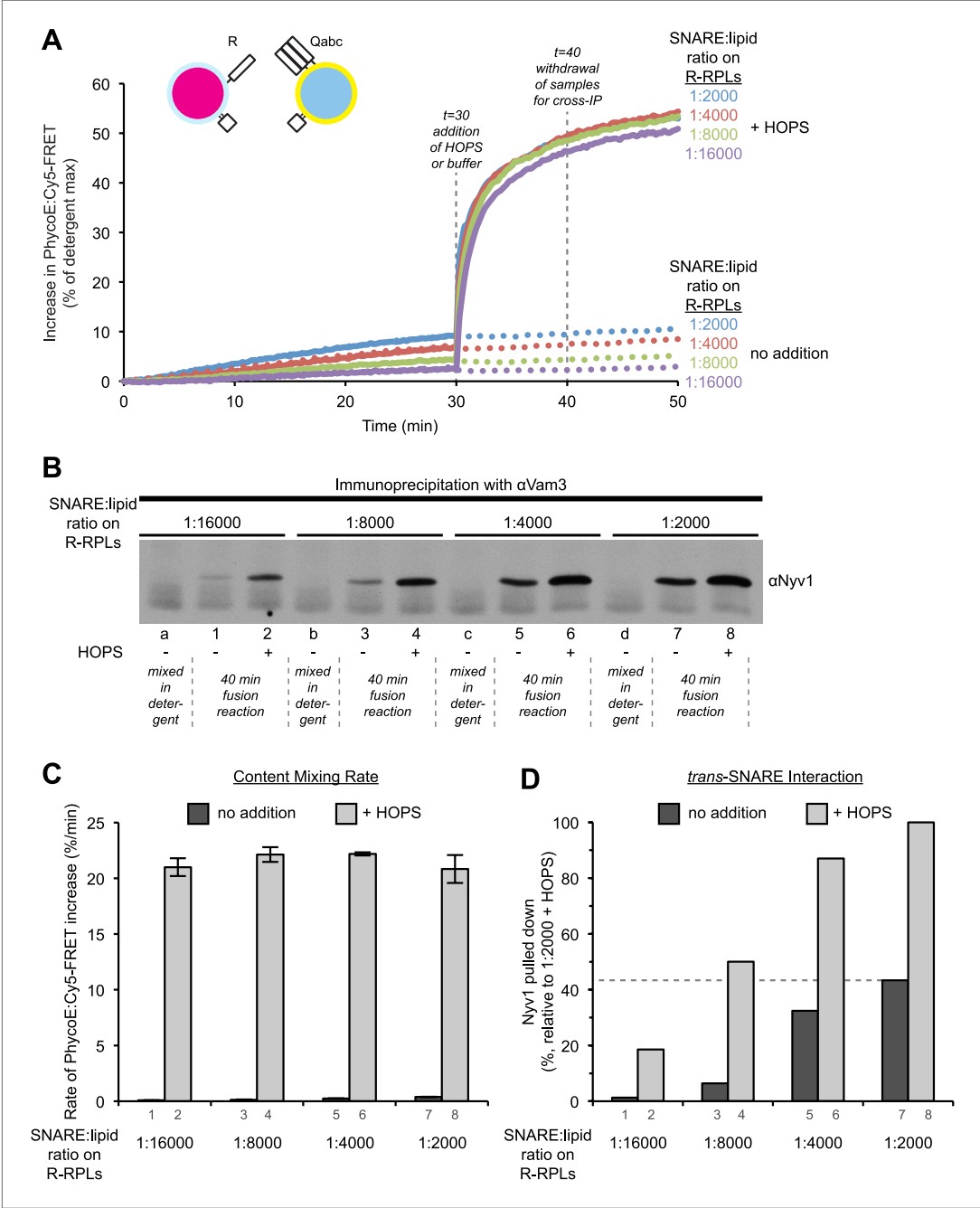

**Figure 7**. High cooperativity does not explain the disproportion between *trans*-SNARE interaction and fusion rate resulting from HOPS addition. (**A**) Kinetics of content mixing reactions of 3Q-RPLs (reconstituted with a protein:lipid ratio of 1:2000 for Vam3p, Vti1p, and Vam7p) and 1R-RPLs (reconstituted with a protein:lipid ratio of either 1:2000 [blue], 1:4000 [red], 1:8000 [green], or 1:16,000 [purple] for Nyv1p). RPLs were incubated for 30 min at 27°C before and HOPS (100 nM) or its buffer were then added. (**B**) Equivalent 40 μl reactions were analyzed for *trans*-SNARE association by co-immunoprecipitation of Nyv1p (R-SNARE) with Vam3p (Q$_a$-SNARE). Lanes 1–8 correspond to the conditions as shown in (**A**) with odd numbers representing conditions with no addition and even numbers representing conditions with HOPS (100 nM). For lanes a, b, c, and d, the components were mixed directly in detergent solution without prior incubation. (**C**) Quantification of content mixing (as seen in panel **A**) from three independent repeat experiments. The rate of fusion was determined as the slope for the first 10 min of the content mixing reaction (between minutes 0 and 10) for samples that did not receive HOPS or as the slope for the first minute of the content mixing reaction after addition of HOPS (between minutes 30 and 31). (**D**) Quantification of *trans*-SNARE association (as seen in panel **B**).
*Figure 7. Continued on next page*

*Figure 7. Continued*

*Trans*-SNARE interactions were quantified as the amount of Nyv1p that co-precipitated with Vam3p (lanes 1–8 in panel **B**) using UN-SCAN-IT gel 5.3 software (Silk Scientific, Orem, UT), and are represented as percentage of the amount of Nyv1p that co-precipitated in the reaction that contained HOPS and R-RPLs with a SNARE:lipid ratio of 1:2000 (lane 8 in panel **B**).

factors are important to produce *trans*-SNARE interaction of the right quality to facilitate membrane fusion. Fusion-incompetent SNARE complexes have been observed in other systems (reviewed in *Brunger, 2005*; *Rizo and Sudhof, 2012*), and it will be crucial to determine what differentiates a productive from a non-productive complex, and what aids the formation of one over the other. Our findings do not vitiate the central role of *trans*-SNARE complexes in fusion; rather, they reveal that a distinct tethering step is a critical upstream event that substantially increases the rate of SNARE-mediated fusion, as suggested (*Smith and Weisshaar, 2011*), and its mode of action is not limited to increasing the quantity of *trans*-SNARE pairings.

Reconstitution studies using lipid dequenching assays and endosomal SNAREs have also suggested that SNAREs alone suffice to drive fusion efficiently. Lipid mixing was reported to occur as a result of fairly promiscuous endosomal *trans*-SNARE pairing (*Brandhorst et al., 2006*) and in multiple topologies (*Zwilling et al., 2007*) without auxiliary proteins. However, another study (*Ohya et al., 2009*), which employed a content mixing assay to measure fusion, found that a Rab GTPase and its effectors are essential for endosomal SNARE-mediated fusion. Neuronal fusion has also been extensively studied through in vitro reconstitution, and led to the concept that SNAREs alone suffice as the machinery of fusion (*Jahn and Fasshauer, 2012*). While SNAREs are undoubtedly essential components, their spontaneous assembly into a four-helical bundle only occurs very slowly (*Pobbati et al., 2006*). Vital components that accelerate this assembly are more recently being included in the definition of a core machinery in this system. Rizo's group has provided compelling evidence for the critical roles of Munc18 and Munc13 during neuronal SNARE-mediated fusion, together with NSF/SNAP and the soluble domain of synaptotagmin 1 (*Ma et al., 2013*), while Brunger's group has shown that complexin has a large effect on $Ca^{2+}$ triggered synchronized fusion in the presence of neuronal SNAREs and full-length synaptotagmin 1 (*Diao et al., 2012*; *Lai et al., 2014*). Further study of these components can broaden our understanding of the complex mechanisms that regulate and facilitate membrane fusion.

Tethering may regulate vacuolar and other fusion systems. The highly selective distribution of each Rab GTPase, which serves as a receptor for its cognate tethering complex (*Hutagalung and Novick, 2011*), could impose specificity on otherwise promiscuous SNARE complex assembly (*Yang et al., 1999*). Membrane repulsion and the conformation of unpaired, membrane-anchored SNAREs may limit the spontaneous formation of *trans*-SNARE complexes. Bringing membranes into close apposition and stabilizing such a tethered intermediate drastically enriches the local concentration of SNAREs and facilitates the interactions that promote *trans*-SNARE complex formation. Recent studies suggested that some multi-subunit tethering complexes could also directly modulate the assembly of functional SNARE complexes (reviewed in *Hong and Lev, 2014*). Nevertheless, even in a tethered state, only a minor fraction (a few percent) of SNAREs seem to engage in such complexes (*Mima et al., 2008*) and lowering the SNARE concentration only a few-fold can take a substantial toll on fusion activity (*Zick et al., 2014*). Other factors that promote fusion, like Sec17p and Sec18p, further stimulate HOPS-dependent reactions (*Figure 1G*; *Mima et al., 2008*; *Mima and Wickner, 2009*), but their mode of action is only poorly understood. The observation that Sec17p/Sec18p relieved the inhibition by soluble SNARE proteins (*Figure 6E* and *Figure 6—figure supplement 3B*) demonstrates the importance of dynamic cycling of SNARE interactions, and highlights the ability of HOPS to selectively protect productive *trans*-SNARE pairs (*Starai et al., 2008*; *Hickey and Wickner, 2010*; *Xu et al., 2010*).

Tethering is integral to our working model of vacuolar homotypic fusion. ATP-dependent priming entails phosphoinositide synthesis (*Mayer et al., 2000*) and the disassembly of *cis*-SNARE complexes by Sec17p/Sec18p/ATP (*Haas and Wickner, 1996*; *Mayer et al., 1996*), providing SNAREs for later assembly of *trans*-SNARE complexes. HOPS mediates tethering (*Hickey and Wickner, 2010*), possibly supported by the *trans*-interaction of the PX domain of Vam7p with PI(3)P (*Xu and Wickner, 2010*). Neuronal SNAREs can associate in multiple conformations (*Weninger et al., 2003*) that can participate in docking (*Bowen et al., 2004*). Similarly, the vacuolar SNAREs may associate in several conformations

when pairing in *trans*. Additional factors that guide the formation of fusion-competent complexes, which might differ from fusion-incompetent complexes in composition, conformation, or associations, are required for rapid fusion. Our data indicate that tethering via HOPS or the binding of the Vam7p PX domain to PI(3)P in *trans* can provide such guidance. Earlier studies of intact vacuoles have also shown that the rate of fusion is not directly proportional to the level of *trans*-SNARE complexes (*Ungermann et al., 1998*), suggesting that only a fraction of *trans*-SNARE interactions might also contribute to fusion activity in vivo.

In addition to its role in facilitating the formation of fusion-competent *trans*-SNARE complexes, tethering may play a direct role in lowering the energy barrier for fusion. Tethering could provide the activation energy required for partially assembled SNARE complexes to fully zipper (*Min et al., 2013*). It could also contribute by inducing membrane bending at the edges of tightly apposed membranes (*Wang et al., 2002*). Domains like the vertex ring in vacuole fusion (*Wang et al., 2002*, *2003*; *Fratti et al., 2004*) require tethering complexes for their formation and stability. Fusogenic lipids, such as the non-bilayer lipids phosphatidylethanolamine, diacylglycerol, and sterol, are required for fusion (*Zick et al., 2014*) and are dependent on other lipids and docking proteins for their enrichment at the vertex ring (*Fratti et al., 2004*). We propose that tethering may thus make several direct contributions to fusion, as well as its role in promoting *trans*-SNARE complexes in conjunction with additional auxiliary proteins.

# Materials and methods

## Proteins and reagents

Lipids were obtained from Avanti Polar Lipids (Alabaster, AL), except for ergosterol which was from Sigma-Aldrich (St. Louis, MO), PI(3)P was from Echelon Biosciences (Salt Lake City, UT), and the fluorescent lipids (Marina-Blue-DHPE, NBD-DHPE, Rhodamine-DHPE, Oregon-Green-488-DHPE, Texas-Red-DHPE) were from Life Technologies (Carlsbad, CA). Biotinylated R-phycoerythrin was purchased from Life Technologies, Cy5-derivatized streptavidin from KPL (Gaithersburg, MD), and unlabeled streptavidin from Thermo Scientific (Waltham, MA). Sec18p (*Haas and Wickner, 1996*), Sec17p (*Schwartz and Merz, 2009*), Ypt7p (*Zick and Wickner, 2013*), HOPS (*Zick and Wickner, 2013*), and vacuolar SNARE proteins (*Mima et al., 2008*; *Schwartz and Merz, 2009*; *Zucchi and Zick, 2011*) were purified as described. Vti1p and Nyv1p were exchanged into octylglucoside buffer as described (*Zucchi and Zick, 2011*). Soluble Nyv1p (sR) was purified as GST-TEVsite-Nyv1(Δtm) as described (*Thorngren et al., 2004*), and cleaved with TEV protease prior to use. GST-His6-3Csite-Nyv1p was purified as described (*Izawa et al., 2012*).

## Reconstitution of vacuolar Rab/SNARE proteoliposomes

Proteoliposomes were prepared by detergent dialysis (20 kDa cutoff membrane) in RB150/Mg$^{2+}$ (20 mM HEPES-NaOH, pH 7.4, 150 mM NaCl, 1 mM MgCl$_2$, 10% glycerol [vol/vol]) as described (*Zick and Wickner, 2013*), with modifications. Lipids dissolved in chloroform were mixed at proportions that mimic the vacuolar lipid composition: 44.6–47.6 mol% POPC (1-palmitoyl-2-oleoyl-sn-glycero-3-phosphocholine), 18 mol% POPE (1-palmitoyl-2-oleoyl-sn-glycero-3-phosphoethanolamine), 18 mol% Soy PI (L-α-phosphatidylinositol), 4.4 mol% POPS (1-palmitoyl-2-oleoyl-sn-glycero-3-phospho-L-serine), 2 mol% POPA (1-palmitoyl-2-oleoyl-sn-glycero-3-phosphate), 1 mol% 16:0 DAG (1,2-dipalmitoyl-sn-glycerol), 8 mol% ergosterol, and 1 mol% (unless specified otherwise) di-C16 PI(3)P (1,2-dipalmitoyl-*sn*-glycero-3-phospho-(1'-myo-inositol-3'-phosphate)). Different fluorescent lipids (Marina-Blue-DHPE, NBD-DHPE, rhodamine-DHPE, Oregon-Green-488-DHPE, Texas-Red-DHPE) were included to allow assays of lipid dequenching or lipid quenching. Concentrations are indicated in the figure legends of the respective experiments. Molar protein:lipid ratios were 1:1000 for SNAREs and 1:2000 for Ypt7p. Isolation after reconstitution was achieved by floatation on a three-step Histodenz gradient (35%, 25% Histodenz [wt/vol], and RB150/Mg$^{2+}$); Histodenz (Sigma-Aldrich) solutions were prepared as 70% stock solution in modified RB150/Mg$^{2+}$ with a reduced concentration (2% [vol/vol]) of glycerol to compensate for the osmotic activity of the density medium; lower concentration solutions were obtained by dilution with RB150/Mg$^{2+}$. The proteoliposomes used in *Figures 6 and 7* were essentially prepared the same way, but with the dilinoleoyl-forms of PC, PE, PS, and PA, and with molar protein:lipid ratios for SNAREs as indicated.

## RPL fusion assays

Fusion reactions of 20 µl were assembled from three pre-mixes: two mixes (5 µl each) of RPLs (250 µM lipid each) in RB150/Mg$^{2+}$ with 5 µM streptavidin (for content mixing reactions only), and one mix

of auxiliary factors (e.g., Sec17p, Sec18p, $Mg^{2+}$:ATP, HOPS, anti-Vam3p, sNyv1p) or their respective buffers. All components were incubated individually at 27°C for 10 min, and then combined in wells of 384-well plates to initiate the reaction. The plates were incubated at 27°C in a fluorescence plate reader for 30–90 min and content and/or lipid mixing signals were recorded at intervals of 5–60 s in a SpectraMax Gemini XPS (Molecular Devices, Sunnyvale, CA) fluorescent plate reader (PhycoE:Cy5-FRET, Ex: 565 nm; Em: 670 nm; cutoff: 630 nm; Marina-Blue quenching or dequenching, Ex: 370 nm; Em: 465 nm; cutoff: 420 nm; NBD dequenching, Ex: 460 nm; Em: 538 nm; cutoff: 515 nm; Oregon-Green dequenching, Ex: 504 nm; Em: 534 nm; cutoff: 515 nm). For content mixing reactions, maximal values were determined after addition of 0.1% (wt/vol) Thesit to samples that had not received streptavidin. To estimate the extent of lipid mixing, liposomes mimicking 100% fusion were prepared by mixing respective 'donor' and 'acceptor' mixed micellar solutions 1:1 prior to liposome formation by dialysis.

## Determination of *trans*-SNARE associations by co-immunoprecipitation

To estimate the amount of *trans*-SNARE association that formed during a reaction, the amount of Nyv1p that co-immunoprecipitated with Vam3p was determined. A 40 µl fusion reaction was incubated for 40 min at 27°C, placed on ice, and diluted 10-fold in RIPA buffer (20 mM HEPES-NaOH, pH 7.4, 150 mM NaCl, 0.2% [wt/vol] bovine serum albumin, 1% [vol/vol] Triton X-100, 1% [wt/vol] sodium cholate, 0.1% [wt/vol] sodium dodecyl sulfate) containing 50 µg/ml of affinity-purified anti-Vam3p antibody and 10 µM GST-Nyv1p. After 20 µl of RIPA-buffer washed protein A magnetic beads (Thermo Scientific, Portsmouth, NH) were added, the mix was incubated while nutating at room temperature for 2 hr. After the beads were washed three times with 1 ml of RIPA buffer, samples were eluted in 100 µl of reducing SDS sample buffer at 95°C for 5 min. Aliquots (20 µl) of each sample were subjected to SDS–PAGE and immunoblotting with anti-Nyv1p antibody.

## Acknowledgements

We are grateful to Dr Joji Mima (Osaka University) for providing the expression plasmid of GST-His6-3Csite-Nyv1p. We thank Amy Orr, Deborah Douville (Geisel School of Medicine at Dartmouth), and Holly Jakubowski (now Duke University Medical Center) for expert assistance.

## Additional information

### Funding

| Funder | Grant reference number | Author |
| --- | --- | --- |
| National Institutes of Health | R01 GM23377-40 | William T Wickner |

The funder had no role in study design, data collection and interpretation, or the decision to submit the work for publication.

### Author contributions

MZ, Conception and design, Acquisition of data, Analysis and interpretation of data, Drafting or revising the article; WTW, Conception and design, Analysis and interpretation of data, Drafting or revising the article

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
