## [Decision Letter]

Thank you for sending your work entitled “A Distinct Tethering Step is Vital for Vacuole Membrane Fusion” for consideration at *eLife.* Your article was handled by Randy Schekman as Senior editor and generally favorably evaluated by 3 reviewers, one of whom is a member of our Board of Reviewing Editors. However, there are some significant concerns that require major revision as outlined below.

The following individuals responsible for the peer review of your submission have agreed to reveal their identity: Axel Brunger (Reviewing editor); Jose Rizo (one of the peer reviewers).

It has been appreciated for some time now that lipid mixing may not necessarily imply content mixing. This work now adds even more rather unsettling evidence that lipid-dye dequenching assays should be treated with caution. The authors find that lipid-dye dequenching is asymmetric in their system, i.e., (de-) quenching occurs much more extensively when lipid dyes are included with one class of vesicles containing R-SNAREs than when they are included with vesicles containing Q-SNAREs. This asymmetry is not due merely to some unusual property of the particular type of fluorescence lipids chosen, since the authors show that analogous results were obtained with three different types of fluorescent lipids. Moreover, the data suggest that the observed dequenching of the lipid dyes is not due to complete fusion. A second key finding is that the tethering of HOPS on the vacuolar SNARE-bearing proteoliposomes initiates efficient lumenal compartment mixing. However, placing PI3P in the R-SNARE liposomes, vacuolar SNAREs alone can also induce efficient content mixing without HOPS.

Major comments:

1) The results raise the question if it is lipid dye transfer that depends on the asymmetric vesicle composition? The authors rightfully say “We do not know how trans-SNARE interactions lead to this dequenching, but it is not symmetric, as expected for true lipid mixing, and is not proportionate to the content mixing for each reaction condition; thus, it cannot be relied on as a faithful reporter of fusion.” What is the origin of the asymmetric lipid dye transfer? The authors should suggest a mechanism, especially since fusion inhibitors suppress the signal. In particular, could this be related to some sort of interaction between some of the vacuolar SNAREs, PI(3)P, and the dyes themselves? In other words, dye-lipids may be transferred to the other membrane that contains a particular SNARE via an interaction with that SNARE? In this context, transfer of lipid dyes between membranes has been previously observed (S. Ohki, T. D. Flanagan, and D. Hoekstra, Biochemistry, 1998, 37, 7496-503) for Rhodamine B lipids. Moreover, is it plausible that positively charged residues in the juxtamembrane regions of the Q SNAREs help to 'extract' the fluorescent lipids from the R-liposomes when the two membranes are brought into proximity by the trans SNARE complexes, thus catalyzing transfer from the R- to the Q-liposomes? Another (albeit, less interesting) possibility that could explain their data is a different size distribution of vesicles and unequal dye partitioning. For example, if the R-SNARE vesicles are much smaller than the Q-SNARE vesicles, then the lipid dyes may partition non-randomly between leaflets in the R-SNARE vesicles but not in the Q-SNARE vesicles (e.g. one could have a much higher concentration of dyes in the outer leaflet of the R-SNARE vesicles than in the Q-SNARE vesicles). Then, upon hemifusion, there would be a greater fluorescence change if the dyes were in the R-SNARE vesicle originally than in the Q-SNARE vesicle. Small amounts of hemi-fusion would appear magnified in that case. Alternatively, there could be some interaction between the R-SNARE proteins and the lipid dye which make it undergo hemifusion more readily than if the dye is on the Q-SNARE vesicles.

2) Relevant to point (1), the authors need to provide some characterization of their proteoliposomes, especially size distribution. This could be using dynamic light scattering or electron microscopy. Also, is there evidence that the protein and lipid dye incorporation is homogeneous?

3) The content mixing assay also shows asymmetry (Figure 1—figure supplement 1, and compare Figure 1, cases with HOPS and Sec18/HOPS). How can this be explained? Since an important aspect of this paper is the asymmetry observed in lipid mixing, it is important to establish whether the differences in the content mixing results constitute a true asymmetry or arise from differences in the amounts of probes trapped in the proteoliposomes in the different configurations. The authors should discuss this point and, ideally, should provide some characterization of the proteoliposomes to distinguish between these possibilities.

4) The effect of asymmetric addition of PI(3)P on vacuolar SNARE-mediated content mixing is also interesting, but perhaps easier to explain. As suggested by the authors, this effect could be explained by a trans interaction between Vamp7p and PI(3)P. In the other vesicle, the effect is due to an increase in “tethering” by this trans interaction. However, upon inclusion of PI(3)P in both vesicle classes does not show such an increase. Could this be explained by “self-interactions” between Vamp7p and PI(3)P in its own membrane?

5) Most of the data (except Figure 5 and Figure 6) only show one time point (fluorescence at ten minutes). Did the authors collect kinetic time traces, i.e. is the 10 min time point a reliable and representative measure in all cases? Or was no kinetic data collected? Because no kinetic data is shown for most figures, the differences in the data could be due to either different rates or different extents of fusion or both. In turn, that might suggest different underlying mechanisms. Showing the kinetic data, even if only in the supporting material, would be very useful. One of the principal advantages of doing bulk phase fusion experiments is that it is much easier and quicker to collect kinetic data, so why not exploit that?

6) The second central claim of the paper is that the tethering action of HOPS is an essential upstream step to the fusion reaction. That may very well be true, but might not the data herein also support the idea that the action of HOPS is merely to stabilize the Q-SNAREs in a fusion-competent configuration such that they can mediate both tethering and fusion? Q-SNAREs in other systems are well known to form dead-end complexes that are not fusion competent (see, for example, reviews by Brunger, A.T. (2006) Quart. Rev. Biophys. 38,1-47. and Rizo, J., and Südhof, T.C. (2012). Annual Review of Cell and Developmental Biology 28, 279-308.). Might that be what is going on here? The authors suggest that only a fraction of their Q-SNARE vesicles seem to be fusion-competent (Figure 2, Figure 5—figure supplement 1, and in the text). Perhaps the essential role of HOPS in their experiments is not to tether the two membranes, but rather to stabilize Q-SNARE complexes in a fusion-competent configuration. Or perhaps both mechanisms of action are essential. The PI(3)P experiments in Figure 5 partially address this question, but as the SNAREs are known to interact with PI(3)P, it may be that PI(3)P also stabilizes the fusion-competent configuration. Outside of single vesicle fusion experiments, which can easily visualize the tethering step, what is really needed is an experiment in which the vesicles are tethered in a manner independent of any of the fusion machinery. In a simplistic way, this could be done by using the same streptavidin tethering experiment as was used to collect the data in Figure 6, but using the yeast SNAREs and Ypt7p instead of the rat SNAREs (see note 7 below). Also, content mixing rather than lipid mixing should be used as a fusion readout in that experiment (see note 5).

7) The data presented in Figure 6 are used as evidence that tethering is also important for membrane fusion induced by endosomal SNAREs. However, all the data shown in the figure rely on a lipid-mixing assay. Hence, reliable conclusions from about membrane fusion cannot be drawn, given all problems inherent to interpret lipid mixing data. It is also unclear why the data are presented as rate of fluorescence decrease, in contrast to other figures of the paper. In addition, the data have large error bars and some reactions with an incomplete group of SNAREs (e.g. 1R 1R) yield a substantial effect that is about 30-40% of that obtained with 3Q and 1R. Therefore, these results are far from demonstrating that tethering is required for endosomal membrane fusion. The data could be omitted as they are not central to the key points of this paper, or a proper content mixing assay used, such as Tb3+-DPA or ANTS-DPX that should work nicely.

8) Another concern with the streptavidin tethering experiments (Figure 6) is the physical role of streptavidin. In some ways, it might seem that tethering with streptavidin would inhibit fusion; many streptavidin molecules might attach themselves to the interface between the tethered vesicles, preventing trans-SNARE binding from that interface and only allowing the SNAREs to bind to each other at the periphery of the contact area. In effect, they might clog up the interface. Is streptavidin added at a low enough concentration that this can be ruled out? What do the authors think is going on here?

[Editors' note: further revisions were requested prior to acceptance, as described below.]

Thank you for sending your work entitled “A Distinct Tethering Step is Vital for Vacuole Membrane Fusion” for consideration at *eLife.* Your article has been favorably evaluated by Randy Schekman (Senior editor) and 3 reviewers, one of whom is a member of our Board of Reviewing Editors. However, there are some remaining issues that need to be addressed.

The following individuals responsible for the peer review of your submission have agreed to reveal their identity: Axel Brunger (Reviewing editor); Stephen Boxer and Joseph Rizo (peer reviewers).

The Reviewing editor and the other reviewers discussed their comments before we reached this decision, and the Reviewing editor has assembled the following comments to help you prepare a revised submission.

We would like to thank the authors for addressing the concerns that were raised in the previous reviews. The new results presented in Figure 6 are indeed very interesting and add to this paper.

Comments:

1) The authors should stress in various places (Abstract, Introduction, and Discussion) that the findings of asymmetric lipid mixing add another problem for lipid mixing assays. Thus, they should not be just “employed with caution” (as stated in the Discussion), but with “great caution”, and should always be accompanied by a complete sets of content mixing experiments (not just one as I have seen in some papers).

2) The authors' interpretation of the new data in Figure 6 (that the role of HOPS in fusion is not just its tethering action), seems to be at odds with the data in Figure 5. In Figure 5, when PI3P is only on the R SNARE vesicles (mimicking the tethering action of HOPS), then similar levels of fusion are observed with and without HOPS. Doesn't this suggest that the central role of HOPS is only to tether the vesicles together? If HOPS is also involved in stabilizing a fusion competent conformation of snare proteins or influencing fusion through some other action, then shouldn't there be an increase in fusion when HOPS is added? The authors should reconcile their interpretation of the data with this point.

3) The authors state that the lipid dequenching signal showed an asymmetric character “suggesting that it arises without the mixing of two bilayers.” We recommend that the authors remove that clause as it seems to suggest a rather unphysical interpretation of the data. The discussion following that statement is sufficient to convey their main point that lipid mixing only experiments should be used with caution.

4) It is good that the authors have included example kinetic traces in the supplementary material. However, upon the inspection of that data, it seems as though the 10 minute time point (which is what is used for the bulk of the data in the main manuscript) is not always a faithful reporter of either the extent or the rate of the fusion reaction. For example, compare the Sec17/Sec18/HOPS trace (purple) to the no addition trace (blue) in Figure 5—figure supplement 1 at t=10 vs. at t=30 min. Why not show the kinetic traces in the main text or represent it in another way?

5) The authors argue that trans-SNARE complexes alone are not sufficient for fusion because some trans SNARE complexes were formed in the absence of HOPS and HOPS increases the amount of trans SNARE complexes three-fold while stimulating fusion 100-fold. Although I think that the authors are most likely right, the results could also be explained if there is high cooperativity in the system so that below a certain threshold in the number of SNARE complexes there is no or little fusion, and fusion becomes much more efficient above this threshold. Perhaps the authors would like to include this in the Discussion.

6) From an editorial perspective, the authors are asked to consider moving most supplemental figures into the main text (except those that are purely “data” figures). The scattering of figures between main figures and supplemental figures makes it more difficult to read the paper. One of the important features of *eLife* is that it does not have a limit on the number of figures, which sets it apart from most other journals. Also, subtitles in the Results section would be desirable to separate some of the key findings.

---

## [Author Response]

*1) The results raise the question if it is lipid dye transfer that depends on the asymmetric vesicle composition? The authors rightfully say “We do not know how trans-SNARE interactions lead to this dequenching, but it is not symmetric, as expected for true lipid mixing, and is not proportionate to the content mixing for each reaction condition; thus, it cannot be relied on as a faithful reporter of fusion.”. What is the origin of the asymmetric lipid dye transfer? The authors should suggest a mechanism, especially since fusion inhibitors suppress the signal. In particular, could this be related to some sort of interaction between some of the vacuolar SNAREs, PI(3)P, and the dyes themselves? In other words, dye-lipids may be transferred to the other membrane that contains a particular SNARE via an interaction with that SNARE? In this context, transfer of lipid dyes between membranes has been previously observed (S. Ohki, T. D. Flanagan, and D. Hoekstra, Biochemistry, 1998, 37, 7496-503) for Rhodamine B lipids. Moreover, is it plausible that positively charged residues in the juxtamembrane regions of the Q SNAREs help to 'extract' the fluorescent lipids from the R-liposomes when the two membranes are brought into proximity by the trans SNARE complexes, thus catalyzing transfer from the R- to the Q-liposomes? Another (albeit, less interesting) possibility that could explain their data is a different size distribution of vesicles and unequal dye partitioning. For example, if the R-SNARE vesicles are much smaller than the Q-SNARE vesicles, then the lipid dyes may partition non-randomly between leaflets in the R-SNARE vesicles but not in the Q-SNARE vesicles (e.g. one could have a much higher concentration of dyes in the outer leaflet of the R-SNARE vesicles than in the Q-SNARE vesicles). Then, upon hemifusion, there would be a greater fluorescence change if the dyes were in the R-SNARE vesicle originally than in the Q-SNARE vesicle. Small amounts of hemi-fusion would appear magnified in that case. Alternatively, there could be some interaction between the R-SNARE proteins and the lipid dye which make it undergo hemifusion more readily than if the dye is on the Q-SNARE vesicles*.

As mentioned under point 2, new data on the characterization of the RPLs has been added. But this data does not support a simple resolution to the mystery as proposed in the 2^nd^ half of this comment. The phenomenon of asymmetry is important in that it uncovered that lipid mixing assays cannot be relied on comfortably. What precise mechanism underlies the observed asymmetry is unclear. We have added a section to the text discussing several possibilities based exactly on the points raised above without drifting too far into speculations.

2) Relevant to point (1), the authors need to provide some characterization of their proteoliposomes, especially size distribution. This could be using dynamic light scattering or electron microscopy. Also, is there evidence that the protein and lipid dye incorporation is homogeneous?

We have added data as Figure 3—figure supplement 2 and Figure 3—figure supplement 3 (DLS size measurements, stained gel, characterization of dye incorporation) to address these points.

*3) The content mixing assay also shows asymmetry (*Figure 1—figure supplement 1*, and compare*
Figure 1*, cases with HOPS and Sec18/HOPS). How can this be explained? Since an important aspect of this paper is the asymmetry observed in lipid mixing, it is important to establish whether the differences in the content mixing results constitute a true asymmetry or arise from differences in the amounts of probes trapped in the proteoliposomes in the different configurations. The authors should discuss this point and, ideally, should provide some characterization of the proteoliposomes to distinguish between these possibilities*.

Under standard reaction conditions no asymmetry was observed for content mixing assays. The differences when comparing Figure 1 arise from the fact that 1D is a 4SNARE-4SNARE reaction, thus depending on Sec17/Sec18 activity to disassemble cis-SNARE complexes, while 1G represents a 1R-3Q reaction, which does not depend on the concerted action of Sec17 and Sec18. The differences that were observed in Figure 5—figure supplement 1 are only seen when the ratio of the two RPL populations is altered. There is no apparent difference when comparing the 1:1 conditions. The differences that can be observed when the ratios are shifted are more likely due to the fact that an imbalance between 1R and 3Q RPLs is created in favor of one or the other. This and other experiments suggest that not all 3Q-RPLs are readily fusion competent. We have modified the text to provide more clarity about these points.

*4) The effect of asymmetric addition of PI(3)P on vacuolar SNARE-mediated content mixing is also interesting, but perhaps easier to explain. As suggested by the authors, this effect could be explained by a trans interaction between Vamp7p and PI(3)P. In the other vesicle, the effect is due to an increase in “tethering” by this trans interaction. However, upon inclusion of PI(3)P in both vesicle classes does not show such an increase. Could this be explained by “self-interactions” between Vamp7p and PI(3)P in its own membrane*?

Precisely this is what we assume to be the case. We have modified the text to make this clear.

*5) Most of the data (except*
Figure 5
*and*
Figure 6*) only show one time point (fluorescence at ten minutes). Did the authors collect kinetic time traces, i.e. is the 10 min time point a reliable and representative measure in all cases? Or was no kinetic data collected? Because no kinetic data is shown for most figures, the differences in the data could be due to either different rates or different extents of fusion or both. In turn, that might suggest different underlying mechanisms. Showing the kinetic data, even if only in the supporting material, would be very useful. One of the principal advantages of doing bulk phase fusion experiments is that it is much easier and quicker to collect kinetic data, so why not exploit that*?

Data has been collected as kinetic time traces. We have now included representative kinetic data as figure supplements for all figures.

*6) The second central claim of the paper is that the tethering action of HOPS is an essential upstream step to the fusion reaction. That may very well be true, but might not the data herein also support the idea that the action of HOPS is merely to stabilize the Q-SNAREs in a fusion-competent configuration such that they can mediate both tethering and fusion? Q-SNAREs in other systems are well known to form dead-end complexes that are not fusion competent (see, for example, reviews by Brunger, A.T. (2006) Quart. Rev. Biophys. 38,1-47. and Rizo, J., and Südhof, T.C. (2012). Annual Review of Cell and Developmental Biology 28, 279-308.)*. *Might that be what is going on here? The authors suggest that only a fraction of their Q-SNARE vesicles seem to be fusion-competent (*Figure 2*,*
Figure 5—figure supplement 1*, and in the text). Perhaps the essential role of HOPS in their experiments is not to tether the two membranes, but rather to stabilize Q-SNARE complexes in a fusion-competent configuration. Or perhaps both mechanisms of action are essential. The PI(3)P experiments in*
Figure 5
*partially address this question, but as the SNAREs are known to interact with PI(3)P, it may be that PI(3)P also stabilizes the fusion-competent configuration. Outside of single vesicle fusion experiments, which can easily visualize the tethering step, what is really needed is an experiment in which the vesicles are tethered in a manner independent of any of the fusion machinery. In a simplistic way, this could be done by using the same streptavidin tethering experiment as was used to collect the data in*
Figure 6*, but using the yeast SNAREs and Ypt7p instead of the rat SNAREs (see note 7 below). Also, content mixing rather than lipid mixing should be used as a fusion readout in that experiment (see note 5).*

The HOPS complex is certainly more than a simple tether, and it remains to be sorted out in detail what “the tethering action of HOPS” involves. Experiments with streptavidin as an artificial tether had been performed in the past (see [65]; Figure 6 – this is lipid mixing data, since our content mixing data also utilizes streptavidin:biotin binding, and small molecule content mixing assays like Tb3+-DPA or ANTS-DPX are impractical due to membrane permeation in our system (see [67])), suggesting that one of the functions is indeed to bring the membranes into close apposition.

We have added new data (new Figure 6), which shows that the 3Q-RPLs readily engage soluble R-SNARE, showing that the partial 3Q complex retains competence for interaction with the R-SNARE. This data also shows that membrane anchored SNAREs engage in trans-associations with little ensuing fusion activity. Adding HOPS increases the amount of trans-SNARE interaction only a few fold, while it increases the rate of fusion much more dramatically (> 100-fold). This suggests that trans-SNARE interactions can be of different qualities, some fusion-competent, some not. Future work will have to delineate the multiple functions of HOPS, and will also provide further insight into what defines a tethering event. We have expanded our discussion of this in the manuscript in conjunction with the newly added data, to address directly the question of whether the 3Q-complex is active and whether or not HOPS might primarily act by stabilizing it, as you suggested.

*7) The data presented in*
Figure 6
*are used as evidence that tethering is also important for membrane fusion induced by endosomal SNAREs. However, all the data shown in the figure rely on a lipid-mixing assay. Hence, reliable conclusions from about membrane fusion cannot be drawn, given all problems inherent to interpret lipid mixing data. It is also unclear why the data are presented as rate of fluorescence decrease, in contrast to other figures of the paper. In addition, the data have large error bars and some reactions with an incomplete group of SNAREs (e.g. 1R 1R) yield a substantial effect that is about 30-40% of that obtained with 3Q and 1R. Therefore, these results are far from demonstrating that tethering is required for endosomal membrane fusion. The data could be omitted as they are not central to the key points of this paper, or a proper content mixing assay used, such as Tb3+-DPA or ANTS-DPX that should work nicely*.

Old Figure 6 has been removed.

*8) Another concern with the streptavidin tethering experiments (*Figure 6*) is the physical role of streptavidin. In some ways, it might seem that tethering with streptavidin would inhibit fusion; many streptavidin molecules might attach themselves to the interface between the tethered vesicles, preventing trans-SNARE binding from that interface and only allowing the SNAREs to bind to each other at the periphery of the contact area. In effect, they might clog up the interface. Is streptavidin added at a low enough concentration that this can be ruled out? What do the authors think is going on here*?

Old Figure 6 has been removed.

*[Editors' note: further revisions were requested prior to acceptance, as described below*.*]*

*1) The authors should stress in various places (Abstract, Introduction, and Discussion) that the findings of asymmetric lipid mixing add another problem for lipid mixing assays. Thus, they should not be just “employed with caution” (as stated in the Discussion), but with “great caution”, and should always be accompanied by a complete sets of content mixing experiments (not just one as I have seen in some papers)*.

We have modified the text to express more clearly that content mixing assays are the current ‘gold standard’ for membrane fusion studies, as several lines of evidence show that conclusions drawn solely from data obtain through lipid mixing assays are not sufficiently reliable.

*2) The authors' interpretation of the new data in*
Figure 6
*(that the role of HOPS in fusion is not just its tethering action), seems to be at odds with the data in*
Figure 5*. In*
Figure 5*, when PI3P is only on the R SNARE vesicles (mimicking the tethering action of HOPS), then similar levels of fusion are observed with and without HOPS. Doesn't this suggest that the central role of HOPS is only to tether the vesicles together? If HOPS is also involved in stabilizing a fusion competent conformation of snare proteins or influencing fusion through some other action, then shouldn't there be an increase in fusion when HOPS is added? The authors should reconcile their interpretation of the data with this point*.

The data in Figure 6 suggests that *trans*-SNARE complexes exist in at least two states, fusion-competent and fusion- incompetent. Future studies will need to determine in what aspects these complexes differ, and what favors the formation of one over the other. At this point, it is not clear what HOPS contributes beyond tethering. Tethering via HOPS might provide a certain spatial arrangement favoring the formation of fusion-competent complexes. The binding of PX:PI3P in *trans* may provide such favorable conditions by other means. We do not necessarily see a contradiction in this. But it is certainly clear that additional work is required to understand the mechanistic details of all events upstream of a fully zippered trans-SNARE complex. We took your comment as a stimulus to expand this portion in the Discussion section.

*3) The authors state that the lipid dequenching signal showed an asymmetric character “suggesting that it arises without the mixing of two bilayers.” We recommend that the authors remove that clause as it seems to suggest a rather unphysical interpretation of the data. The discussion following that statement is sufficient to convey their main point that lipid mixing only experiments should be used with caution*.

Removed.

*4) It is good that the authors have included example kinetic traces in the supplementary material. However, upon the inspection of that data, it seems as though the 10 minute time point (which is what is used for the bulk of the data in the main manuscript) is not always a faithful reporter of either the extent or the rate of the fusion reaction. For example, compare the Sec17/Sec18/HOPS trace (purple) to the no addition trace (blue) in*
Figure 5—figure supplement 1
*at t=10 vs. at t=30 min. Why not show the kinetic traces in the main text or represent it in another way*?

We have consciously chosen to present the 10 minute time point, rather than the full kinetic traces, in the main manuscript. It aids the comparison of different conditions (making it more easily accessible to a broader readership) and provides an adequate measure in almost all cases. The kinetic traces are provided for expert readers, and we now explicitly point out the two conditions which exhibit a distinct pattern (i.e., conditions with Sec17/Sec18 and HOPS with no PI3P on the 3Q-RPLs – conditions under which there is no apparent synergy between Sec17/Sec18 and HOPS).

*5) The authors argue that trans-SNARE complexes alone are not sufficient for fusion because some trans SNARE complexes were formed in the absence of HOPS and HOPS increases the amount of trans SNARE complexes three-fold while stimulating fusion 100-fold. Although I think that the authors are most likely right, the results could also be explained if there is high cooperativity in the system so that below a certain threshold in the number of SNARE complexes there is no or little fusion, and fusion becomes much more efficient above this threshold. Perhaps the authors would like to include this in the Discussion*.

We have experimentally addressed the possibility of high cooperativity as a reason for our observations, and present data that rules out this possibility in new Figure 7. In short, lowering the protein:lipid ratio on R-RPLs (while keeping it constant on the 3Q-RPLs) does not substantially affect the amount of fusion observed in the presence of HOPS, while the amount of *trans*-SNARE interactions decreases. The amount of *trans*-interaction at a protein:lipid ratio of 1:16000 on R- RPLs in the presence of HOPS is lower than at a protein:lipid ratio of 1:2000 on R-RPLs in the absence of HOPS, yet the maximal rate of fusion observed under these two conditions is vastly higher for R-RPLs at 1:16000 with HOPS than for R-RPLs at 1:2000 without HOPS (compare Figure 7, panels C&D, columns 2&7).

*6) From an editorial perspective, the authors are asked to consider moving most supplemental figures into the main text (except those that are purely ”data” figures). The scattering of figures between main figures and supplemental figures makes it more difficult to read the paper. One of the important features of eLife is that it does not have a limit on the number of figures which sets it apart from most other journals. Also, subtitles in the Results section would be desirable to separate some of the key findings*.

We have carefully considered both requests. Subtitles were added to the Results section. Regarding the Supplementary material, we have chosen to leave the arrangement as it was. We agree that supplementary figures sometimes disrupt the flow of reading a paper, but also feel that the same happens when there are too many “primary” figures. In our opinion, *eLife* has an elegant solution by showing additional data as Figure supplements rather than Supplementary figures. What initially might appear to be a matter of semantics, is in fact an elegant twist that becomes apparent when reading the online version of an *eLife* paper. All Figure supplements are readily accessible right next to the corresponding primary figure. We find that this is indeed the best solution, and hope that you agree.